# LayoutRL: A Reinforcement Learning-Based Approach to Keyboard Layout Optimization

## Abstract

Keyboards are a key interface between humans and computers, with character arrangements offering numerous layout possibilities. Many existing designs follow standardized ergonomic principles and explore Pareto-optimality in multi-objective functions using metaheuristics or deep learning. In this work, we propose a reinforcement learning-based approach to designing optimized keyboard layouts that integrate both technical and ergonomic considerations. Our results demonstrate that reinforcement learning optimization can produce layouts more efficiently than conventional designs, such as the "QWERTY" keyboard. Specifically, our approach achieves approximately an 12.4% improvement in ergonomic parameters over traditional keyboards, underscoring the potential for a more data-driven, systematic approach to keyboard layout optimization.

## 1 Introduction

keyboard is an essential device that is used to interact with computer systems. Different types of keyboards exist, such as physical keyboards we interface with computers and laptops (Lugay et al., 2022) as well as soft keyboards (Han & Kim, 2015) such as touchscreen-based keyboards, virtual keyboards, etc (Fennedy et al., 2022; Lee et al., 2022). The layout, i.e., the arrangement of characters in these keyboards, may vary depending on different layout models and factors. Typing efficiency also differs according to different keyboard layouts (Deshwal & Deb, 2006). A poor arrangement of characters in the keyboard layout may put a high load on weaker fingers along with typing discomfort resulting in fatigue for the users (Deshwal & Deb, 2006). Hence, optimized arrangement of the characters in any type, language-based keyboard is necessary for comfortable as well as efficient writing. In English keyboard layouts, 'QWERTY' is the most established and familiar layout so far (Khan & Deb, 2023), proposed by the Sholes brothers in 1873 (Eggers et al., 2003). However, several researches have shown that 'QWERTY' is not an optimized layout. Khan & Deb (2023) and Oladeinde et al. have discussed its shortcomings in details. Oladeinde et al. shows that novice typists make fewer errors in Dvorak and alphabetic keyboards compared to 'QWERTY' as well as the efficiency in these layouts can reach the level of 'QWERTY' if their usage is continued in novice typists. However, the popularity and usage of 'QWERTY' has sustained all other optimized layouts till now. Despite that optimization of keyboard layout is an active research field where the fundamental objective is usually to optimize the efficiency of typing (T.G. et al., 2018; Ghosh et al., 2011; Khan & Deb, 2023). This is necessary as the typing efficiency will decrease if the frequent characters are not found within the accessible range.

A vast amount of research has been done on optimizing physical keyboard layout considering different approaches while applying various optimization algorithms such as genetic algorithms, simulated annealing, Ant colony Optimization (ACO), etc, (Glover, 1987; Light & Anderson, 1993; Oommen et al., 1989; Liao & Choe, 2013; Khorshid et al., 2010). Moreover, the improvement on the soft keyboard's layout has also been investigated as suggested by the works of Lewis et al.; MacKenzie & Zhang (1999); Zhai et al. (2000). Focus has been given to both normal and ambiguous keyboards (Deshwal & Deb, 2006). In the case of normal keyboards, importance has been given to mapping the respective language's characters on the keys efficiently, resulting in ease of use for the typists and typing efficiency. Different ergonomic designs and their optimization have been explored in ambiguous keyboards (Deshwal & Deb, 2006; Lesher et al., 1998). In recent studies, emphasis has been given to optimizing layouts in different soft keyboards also, such as virtual keyboards (Ghosh et al., 2011), touchscreen keyboards as well as gesture-based typing (Dunlop &

Levine, 2012; Bi et al., 2014; Smith et al., 2015; Oulasvirta et al., 2013), single-finger keyboard layouts on smartphones (Herthel & Subramanian, 2020; Turner et al., 2020), etc. While optimizing the keyboard layouts in previous works, many articles focused on a single objective function for optimization. This approach can find an efficient keyboard but usually results in very dissimilar to the 'QWERTY' layout as indicated by the works of Eggers et al. (2003); Liao & Choe (2013); Khorshid et al. (2010). Such layouts while efficient can be difficult for users to track and learn compared to conventional 'QWERTY' layouts. To solve this issue, Khan & Deb (2023) considered a multi-objective optimization approach while generating efficient layouts. They focused on two objective functions: maximizing typing efficiency by considering the cumulative distance between consecutive keystrokes while typing large texts and another is maximizing similarity to 'QWERTY'. Khan & Deb (2023) applied the NSGA-II algorithm to produce a Pareto set of layouts ranging from highly optimized ones to more similar to 'QWERTY' while being less optimized ones. Bi et al. (2010) used the metropolis energy minimization algorithm to generate a quasi-qwerty and optimized version of soft keyboards. They found that characters moving one key away from their key positions compared to the 'QWERTY' achieved efficient layout as well as remained similar to 'QWERTY'. Nivasch & Azaria (2021) proposed a deep learning model with a genetic algorithm to optimize for layout of the keyboard. Similar works such as Bi & Zhai (2016); Zhai & Kristensson (2008) conducted optimization of 'QWERTY' while maintaining similarity to it.

However, while designing and optimizing the keyboard layout, proper ergonomic criteria need to be considered. These criteria should be properly mathematically modeled and validated considering the user's interactions with the keyboards, typing comfort, etc. Ergonomic criteria can be a collection of heuristic rules derived from experimental studies and human-keyboard interactions such as rules proposed by Marsan (1976). These sets of rules can work as multiple objectives that need to be optimized to get an efficient layout. Eggers et al. (2003) was one of the first to consider such sets of ergonomic criteria provided by Marc Oliver Wagner & Eggers (2003) and applied the Ant colony optimization algorithm to develop an optimized and ergonomic-supported keyboard layout. However, the ergonomics criteria are not straightforward, and achieving optimal layouts utilizing such criteria is still an active research field. Previous optimization algorithms have utilized different ergonomic criteria but further optimization needs to be explored (Eggers et al., 2003; Khan & Deb, 2023; Bi et al., 2010). Conversely, these ergonomic criteria are not always validated in previous works and need to be addressed from a technical perspective. Furthermore, many works focused on optimizing layout only considering a combinational problem. However, keeping similarities to the 'QWERTY' layout as well as the optimized version is one of the recent trends in keyboard layout optimization research as discussed above.

To address these aforementioned challenges, we propose a novel keyboard layout optimization method called, 'LayoutRL' that uses Reinforcement Learning (RL) to achieve high optimality in the ergonomics criteria. We consider the six ergonomics criteria from Eggers et al. (2003); Marc Oliver Wagner & Eggers (2003) while using an RL algorithm to design an optimal keyboard layout. RL has the ability to find the optimal conditions of all the objective functions while generating the desired system. Additionally, we show that LayoutRL can achieve an optimized yet similar layout to the 'QWERTY. Furthermore, we explored the contribution of these six ergonomic criteria and discussed their validity in constructing an optimized keyboard layout.

## 2 METHODOLOGY

LayoutRL works on the standard 'QWERTY' English keyboards and learns the policy to converge on the optimized layout, producing more efficient layouts than 'QWERTY'. The RL approach considers six ergonomic criteria to learn the optimal policy. Additionally, we have added constraints so that it remains similar to the 'QWERTY'. In this section, we first discuss the keyboard structure and the ergonomic criteria considered. Then, we discuss how the RL model can be formulated to optimize the layout. Finally, we discuss the LayoutRL optimization process and its algorithm to understand its capabilities further.

### 2.1 KEYBOARD STRUCTURE AND ERGONOMICS CRITERIA CONSIDERED

We consider a typical 'QWERTY' keyboard layout where the alphabets are mapped to each key. These keys are divided into columns of rows. A sample of the keyboard layout we considered is

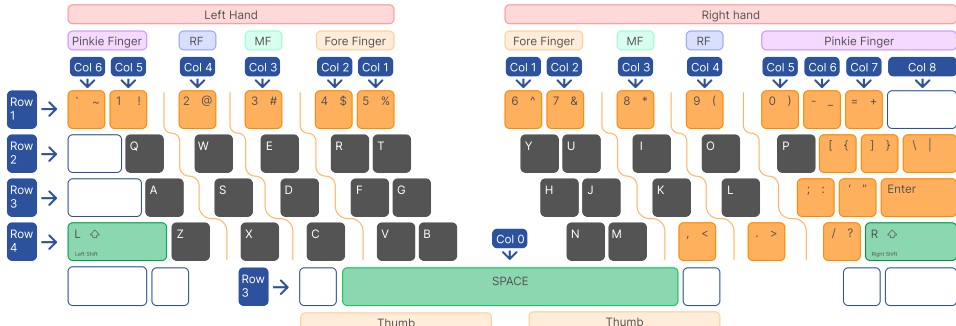

Figure 1: Representation of our keyboard layout structure for the standard "QWERTY" keyboard layout.

shown in Figure 1. A similar layout was mentioned in this work by Eggers et al. (2003). Like this work, We also consider the following structures for our keyboard layout:

- a hand (left, right),

- a column (0–7 for the left hand and 0–8 for the right hand),

- a row (0–5, 0 standing for the top row and 3 for the rest row).

Accordingly, each key in this layout can be mapped to 5 different values: Hand, Row, Column, Finger, Shift. We represent this mapping of the key to the following, **Key** : **Hand**[1 for left and 2 for right], **Row** [0 to 5], **Column** [0 to 7 or 0 to 8], **Finger**[thumb-0,forefinger-1,middlefinger-2,ringfinger-3,pinkie-4], **Shift**[1 for shift else 0]

The objective function for our keyboard layout was formulated according to the six ergonomic criteria mentioned in the research work by Eggers et al. (2003):

- `Accessibility and load`: Distribution of the keys should be such that the load is shared equally by all the fingers for an optimal layout. The cost associated with this criteria can be formulated using the following equation 1 (Eggers et al., 2003):

$$c_1 = \sum_{m_i \in \Xi_1^m} \left( f_{m_i} - f_{m_i}^{\text{opt}} \right)^2 \tag{1}$$

  $f_{m_i}$ indicates the frequency of a monograph. Monograph indicates an isolated key that is struck in the process of typing a text.

- `Key Number`: While typing out the text, the number of hits (keystrokes) on the keyboard should be minimized which can be considered as the second ergonomic criterion. For a layout generated from 'LayoutRL', we run a large corpus using the generated layout and count the number of keystrokes needed to type out the corpus. This cost is denoted by $c_2$.

- `Hand Alteration`: Consecutive hits of the keys should be pressed by alternate hands while typing out large texts. The cost associated with this criteria can be calculated using the following equation (Eggers et al., 2003):

$$c_3 = \sum_{d_i \in \Xi_3^d} f_{d_i} \tag{2}$$

  In Equation.2, $f_{d_i}$ indicates the frequency of a digraph. Digraph indicates the occurrence of two consecutive keys being hit in the process of typing a text. Consecutive keys should be pressed by alternating hands of typists for the ease and comfort of typing. In our cost calculation algorithm, we track the consecutive keys being pressed by the alternating hands or not. The cost is proportional to the number of times consecutive keys that are pressed by the same hands.

- `Consecutive usage of the same finger`: Similarly, consecutive hits of the keys should be pressed by different fingers rather than the same finger. The cost associated with this criteria can be calculated using the following equation 3 (Eggers et al., 2003):

$$c_3 = \sum_{d_i \in \Xi_4^d} f_{d_i} dist(d_i) \tag{3}$$

In equation 3, when typing digraphs, the distance between the consecutive keys is calculated and multiplied. The larger distance would result in a higher cost, hence multiplied. The distance is calculated using the proposed Manhattan distance in equation 4:

$$dist(d_i) = |c_2 - c_1| + |r_2 - r_1| \tag{4}$$

Here, the $c_1$, $r_1$ and $c_2$, $r_2$ are the respective consecutive keys column and row values.

- `Avoid big steps`: Awkward hand positions such as big steps while typing out consecutive characters should be avoided. The cost associated with this criteria can be calculated using the following equation 5 (Eggers et al., 2003):

$$c_5 = \sum_{d_i \in \Xi_5^d} k(d_i) f_{d_i} \tag{5}$$

$k(d_i)$ is the big step coefficient that can emulate the awkward positions and big steps between different fingers. Higher coefficients indicate big steps. Hence, we conclude that the cost in this criterion is proportional to $c_5$.

- `Hit direction`: The optimal keyboard layout should be such that while typing out consecutive words, the small fingers are used first and then gradually towards the thumbs. The cost associated with this criteria can be calculated using the following equation 6 (Eggers et al., 2003):

$$c_6 = \sum_{d_i \in \Xi_6^d} f_{d_i} \tag{6}$$

According to Eggers et al. (2003), the preferred hit direction in consecutive characters is from little fingers towards the thumb. We discussed how the fingers are mapped in section 2.1. In this case, We track the consecutive keywords and the respective fingers that are used while typing to determine the undesirable hit directions. Hence, we can conclude that the cost, $c_6$ is proportional to the frequency of undesirable hit directions.

The weighted average of these criteria are used as total cost for a particular keyboard layout. The details of the weighting mechanism is adopted from Eggers et al. (2003).

## 2.2 DESIGN CHOICES

While optimizing the layout using our proposed method, we set some design constraints while using RL to keep similarity with the 'QWERTY' layout and the best layout generated from 'LayoutRL'. Though this constraint limits high-level optimization of the state space freely, it has certain appeals due to the general acceptance of the 'QWERTY' to mass keyboard users. Our model maintains the following constrains in the design phase:

- `Restrictions on the placement of alphabet and non-alphabet characters`: In the 'QWERTY' layout, the English alphabets and non-alphabet numbers are placed in separate spaces by grouping. We enforce our algorithm to do the same by putting restrictions in state space while assigning them a character. For example, the non-alphabet characters can be placed only in the orange-colored keys only, as specified in Figure 1. Conversely, the alphabet characters can be assigned only on the black-colored keys as demonstrated in Figure 1. This way we try to ensure more similarities between our best layouts with the 'QWERTY' layout.

- `Static placement of Shift, Alt, and Space button`: We considered fixed positions of Shift, Alt, and Space buttons on the same keys 'QWERTY' has and do not include in the optimization algorithm.

- `Dynamic stroke of both handed buttons for the hand-alteration`: Furthermore, we consider the shift, Alt, and Space buttons on both left and right hand in the layout. This is to ensure for consecutive key pressing, alternating keys are pressed for proper hand alteration as discussed in the 'Hand Alteration' ergonomic criteria.

## 2.3 MARKOV DECISION PROCESS FOR LAYOUTRL

We utilized a Markov decision process (MDP) formulation for optimizing the 'QWERTY' layout. Following are the components of the Markov Decision Process (MDP) model for LayoutRL.

- `State Space`: We consider each possible alphabet-key pair, along with all other assigned and unassigned keys on the keyboard constitute the states, $S$. There are 95 unique characters, including capital and small alphabets and special characters, allocated to 96 keys. The possible state space can have approximately $^{96}P_{95}$ different states. Due to this large state space, it is computationally unfeasible to check every possible combination to find the best solution. We consider the 'shift + key' as separate keys on the keyboard.

- `Actions`: An action, $a \in A$ in our MDP problem is defined as mapping an alphabet to one of the 96 keys. While mapping, the algorithm checks whether the chosen key is already occupied.

- `Episode`: Each episode, $E$ consists of 95 actions taken by the agent, resulting in a complete keyboard layout with the specified number of alphabet-key pairs. After each episode, the generated keyboard is evaluated based on the six ergonomic criteria discussed above, and a score is saved for use in subsequent episodes.

- `Agent`: The agent in this context creates a keyboard layout based on the available keys and alphabets. The decision to create an alphabet-key pair is based on two factors: the availability of the key and the weight associated with the key. To maximize the reward, the agent selects the key with the highest value for a specific alphabet.

- `Reward` ($r$): For each action, $a$, the agent attempts to create an alphabet-key pair with the highest weight. Initially, the reward for every pair is considered uniform, leading to random allocations. Over several episodes, the state/action value matrix (weight matrix) is updated after evaluating the return for each of the generated keyboards.

- `Return`: At the end of each episode, the performance of the newly generated keyboard is evaluated by typing out a standard corpus and considering 6 ergonomic criteria. This evaluation compares the typing cost of the new keyboard against a set number of previous episodes. The return is determined by the rate of improvement in the new layout and can be both positive and negative. The return also influences changes in the state/action value matrix for that specific layout.

- `State transition probabilities` ($p(s, r|s, a)$): The state transition in this model is non-deterministic. The probability of state transition is determined by the state/action value matrix, where the weights of key allocation for an alphabet are evaluated, and the key with the highest weight/probability is chosen.

- `Discount Factor` ($\gamma$): The agent takes actions by assigning a key to an alphabet and moves to another alphabet for assigning. However, since we formulate the policy to update based on the total return after a complete episode, we conclude that state changes have no significant benefits and set the discount factor, $\gamma$ to zero.

- `Policy` ($\pi$): The ultimate objective is to learn the optimal policy ($\pi^*$) that will determine the optimal mapping of alphabets to different states. We learn the policy for our MDP from the total cost of the layout after each episode. Initially, a random/equiprobable policy is used for mapping alphabets to keys. After each episode, we compare the total cost of the present layout to the previous optimized layout. If the total cost reduces, we treat the present layout as the optimized one and increase the corresponding state/action value. Conversely, we decrease the state/action value if the total cost increases. The state/action value will eventually converge, providing the optimal policy.

The value function in the states can be calculated according to the following definitions (Andrew & Richard S, 2018):

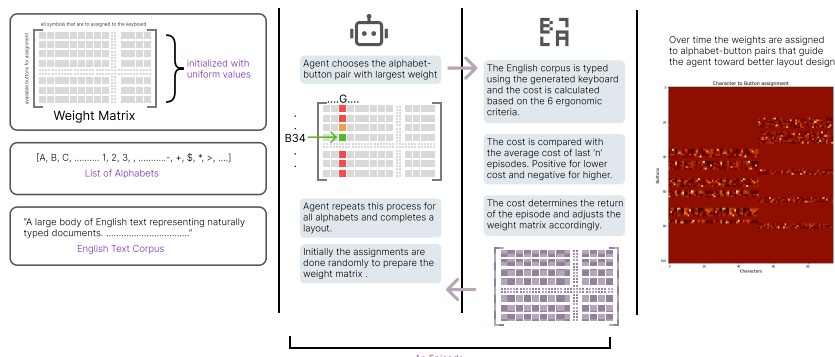

Figure 2: MDP Approach to obtain optimal layout for the English keyboard

$$v_\pi(s) = \sum_{a \in A} \pi(a|s) \sum_{s' \in S} \sum_{r \in R} p(s', r|s, a) [r + \gamma v_\pi(s')] \quad \text{for all } s \in S \tag{7}$$

According to equation 7, the proposed LayoutRL starts with the equiprobability of assigning an alphabet to a key. Then, each complete episode consists of assigning all the alphabet to keys, and after generating a completely new layout, the expected cost is calculated which is evaluated, $r$. If the cost is higher, LayoutRL considers it a negative reward system and updates the state-action matrix, $v_\pi(s|a)$ as well as the policy, $\pi(s|a)$. We can observe the overall formulation of how we approached reinforcement learning-based solutions for our keyboard problem in Figure 2.

In Figure 2, we initially took the 'QWERTY' layout as the 'base' layout and determined its total cost using the objective function. Additionally, we initialize a state/action value matrix where the rows are the available keys on the keyboard and columns represent the alphabets and symbols. Initially, each alphabet has an equal probability of being assigned to any of the keys. Then, we start assigning the alphabet to the empty keys. For English alphabets, When the small character is assigned to one key, the corresponding capital character is also simultaneously assigned to the 'shift + key'. For non-alphabet characters, no such restriction is imposed on the assignment of a 'key' or 'shift + key.' After assigning all the alphabets to the keys, the model types out the corpus using the new layout generated from the algorithm. The total cost of the keyboard is determined and compared with the total cost of the 'QWERTY' layout. If the total cost becomes lower than the 'QWERTY' layout, we consider that layout to be the optimized one compared to the 'QWERTY' layout. This improved cost is used to increase the weights within the state/action value matrix for 'alphabet-key' pairs that have been used in the generated keyboard, simultaneously reducing the weights for others. On the other hand, if the total cost of the layout is greater than the 'QWERTY' layout, then the next episode of the MDP to update the state/action value matrix. For updating, we add a negative value to the already selected alphabet-key pair as the total cost of the layout is high and increase the other values in the matrix. This way, we can converge the state/action matrix to determine the optimal policy and get an optimized layout then the 'QWERTY' layout. Furthermore, we can set the base layout as the newly generated optimized layout and continue repeating the episodes to find the more optimized layout.

### 2.4 PROPOSED OPTIMAL KEYBOARD DESIGN ALGORITHM

The overall optimization process in LayoutRL's algorithm can be divided into two events: random episodes for initial policy learning and optimizing episodes for generating the optimized layouts. The function $Run\_Series$ function initializes the state-action matrix for the keyboard and then random episodes are run to make the state-action matrix randomized for effective convergence in the optimal policy learning. In the optimizing episodes, the agent assigns all the characters to keys based on the weights(probabilities) in the randomized state-action matrix.

After finishing the episode, the $run\_episode$ function returns the cost for the newly generated layout from 'LayoutRL'. Based on this returned cost compared with the previous average cost, the state-action matrix is updated. LayoutRL iteratively runs and checks the cost ratio comparisons until

**Algorithm 1:** Pseudo code for the LayoutRL

**Input:** Number_of_random_episodes, Number_of_optimizing_episodes, Learning_rate, Cost_for_QWERTY, Triviality.
**Output:** Cost_for_the_best_keyboard, Best_keyboard_layout, State/Action_value_matrix.
**Data:** English_corpus, Set_of_symbols, Accessible_buttons.

```
1
2  Run_Series(input,data)
3
4  Function Run_Series(Inputs, Data):
5      State/Action_value_matrix = uniform_matrix of shape(Number_of_keys X Number_of_alphabets)
6      for episode in Number_of_random_episodes do
7          keyboard, cost = run_episode()
8          improve_amount = ((avg_previous_costs - cost) / avg_previous_costs) * Learning_rate
           /* The average costs of the generated layouts are recorded                          */
9          avg_previous_costs = (avg_previous_costs X episode + cost) / (episode + 1)
10         adjust_corresponding_values(State/Action_value_matrix, improve_amount, keyboard)
11     for episode in Number_of_optimizing_episodes do
12         keyboard, cost = run_episode()
13         improve_amount = ((avg_previous_costs - cost) / avg_previous_costs) * Learning_rate
           /* weighted average costs of the generated layouts are recorded                     */
14         avg_previous_costs = ( avg_previous_costs * (Triviality - 1) + cost) / Triviality
15         adjust_corresponding_values(State/Action_value_matrix, improve_amount, keyboard)
16         if cost ¡ best_recorded_cost then
17             best_recorded_cost = cost
18             best_keyboard = keyboard
19     return best_recorded_cost, best_keyboard, State/Action_value_matrix
20
21  Function run_episode(Inputs, Data):
22     for alphabets in Set_of_symbols do
23         if random_assignment_episode then
24             keyboard = create random key-alphabet pairs
25         else
26             keyboard = create key-alphabet pairs with highest weights in State/Action_value_matrix
27         cost = type_the_corpus_with_ergonomics(keyboard, English_corpus)
28     return keyboard, cost
29
```

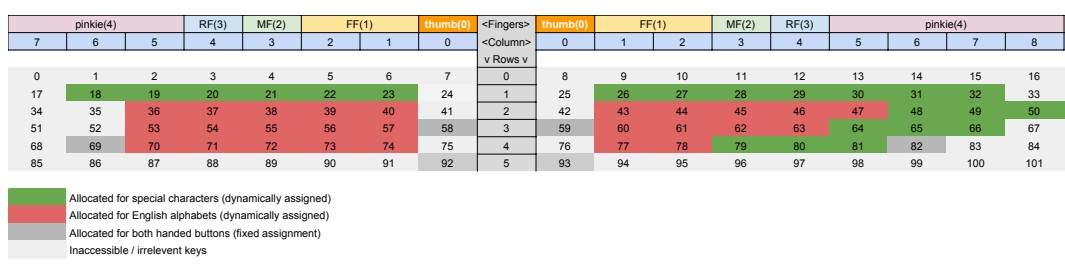

Figure 3: Representation of the metadata provided by the generated keyboards

the cost convergence. Different keyboard layouts with lower costs can be generated by running LayoutRL's algorithm.

## 3 EVALUATION

After the $Run\_Series$ function is completed, several artifacts are produced that can be utilized to evaluate the generated layout design. The artifacts are the best keyboard design, the cost of using that keyboard to type the corpus, the state/action value matrix, and a series of costs generated by each episode. In the following sections, each of the artifacts is evaluated and analyzed.

### 3.1 ARTIFACT: BEST KEYBOARD

This artifact is generated as a dictionary, where the keys represent the alphabets/symbols and the values identify the corresponding button number along with other metadata like which hand is used for the stroke, row number, column number, which finger is used for the stroke, if shift is used, and

| Keyboard | Uses traditional layout structure | Cost of corpus completion | comparative cost |
|---|---|---|---|
| QWERTY | Yes | 37373 | 100% |
| AZERTY (Eggers et al., 2003) | Yes | 38618 | 103.3% |
| Colemak (Francis, 2015) | Yes | 36210 | 96.9% |
| Ant-keyboard (Eggers et al., 2003) | No | 30422 | 81.4% |
| **LayoutRL** | Yes | 32765 | 87.6% |

Table 1: Cost performance comparison with best designs from LayoutRL with other standard keyboard layouts

the button number. This lets us reconstruct the physical keyboard layouts. Figure 3 provides a visual representation of the metadata provided by the dictionary.

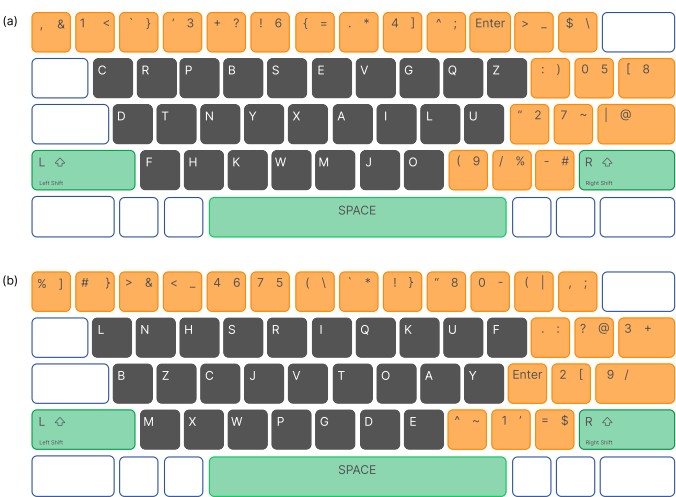

Figure 4: Samples of Keyboards generated by 'LayoutRL' with low cost compared to the 'QWERTY' Layout (a) Best Layout 1 (b) Best Layout 2 from 'LayoutRL'.

Figure 3 demonstrates the numbering of all 102 keys in the layout. LayoutRL returns the best keyboard layout in a dictionary containing the hand, row, and column which can be mapped to the key locations. For example, key 38 indicates that this key is on the left side of the layout having been placed exactly at the 2nd row and 3rd column.

### 3.2 ARTIFACT: COST OF USING THE BEST KEYBOARD DESIGNS AND STATE/ACTION VALUE MATRIX

As discussed and shown in section 2.2 and Figure 3, our design keeps the generated layout similar to 'QWERTY'; the generated keyboard layouts follow the traditional keyboards in the placement of the alphabet and special characters in their separate sections. Figure 4 shows two such layouts with 12% and 12.5% lower cost scores than the QWERTY keyboard. We can see the cost for each of the six ergonomic criteria for the second layout in table 2. Though the table does not perform exceptionally in any particular aspects other than 'Big steps', the cumulative cost is lower than all the traditional layouts. As the layout is not performing highly in any particular area, specific design criteria or patterns are hard to identify in the design. We compare the cost of the best designs from 'LayoutRL' with other standard layouts as demonstrated by table 1. This table shows that the cost determined according to the six ergonomic criteria, LayoutRL beats every standard layout except the Ant-keyboard layout. However, the lower cost of the Ant-keyboard is due to the free mix of the alphabet and non-alphabet characters, resulting in high optimization of the space. Yet, it is not a traditional layout structure and required special button placement in the physical layout. From table 1, we conclude that our best designs adhere to the traditional keyboard structure while reducing the typing cost making them feasible to use with regular keyboard with custom keybinding software. This removes the hurdles of designing and manufacturing new products. Additionally, users can explore these layouts without any minimal cost.

Moreover, The state/Action matrix is another artifact in LayoutRL's results. It is initially uniform while policy learning, and evolves with every episode. Figure 5 shows how the values are adjusted over time. After initiation with uniform values, the first episode creates a keyboard layout and compares its cost against the base cost. If the new cost is lower than the base cost the alphabet-key pairs within the state/action matrix are set to a higher value than the rest which can be seen in the first image in figure 5. Here, the matrix colors are black and white representing lower weight for the unassigned buttons and higher value for the assigned ones respectively. Over time different alphabet-keys are tested in different layouts per episode. The new costs per episode are compared against the weighted average of previous layouts, leading to the evolution of the state/action matrix as shown in the later three images in figure 5. We can observe a converging pattern in the matrix over time.

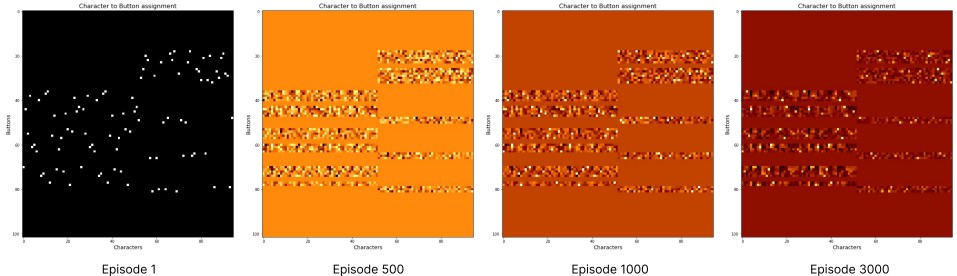

Figure 5: The evolution of the state/action value matrix.

### 3.3 ARTIFACT: SERIES OF COST/EPISODE

The last artifact helps us understand how the costs of the layout design adjust over time. In figure 6 for episodes 0-2400, the episodes are run with random assignments. This helps us prepare the state/value matrix for optimization in the later episodes. In the first 2400 episodes, we can see that there is no converging pattern in the costs of the generated layout as expected from a random assignment strategy.

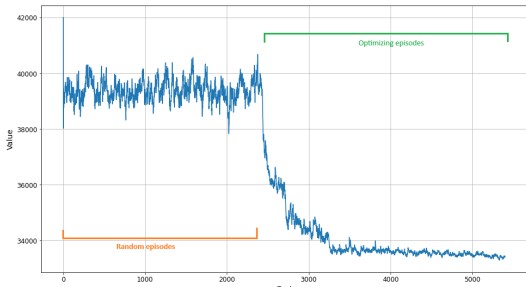

Figure 6: The evolution of the state/action value matrix. Hyperparameters (Preparatory episodes = 2400, Optimizing episodes = 3000, Learning rate = 0.4, Triviality = 10)

After 2400 episodes of random assignments, the optimization episodes start where we observe a sharp decline in the costs of the generated layout according to the figure 6. This is due to the policy defined for the assignments of alphabet-key pairs. It indicates the improvement in the generated layout for the optimizing episodes. The cost values converge near the cost of 33000 in this particular case. As the arrangements are initiated randomly for the primary episodes, the resulting keyboard and minimum cost at the end of each series can differ to some extent, as seen in figure 4.

In addition to the artifacts discussed above, our methodology allows us to determine which keyboard layout performs best among each of the six ergonomic criteria. In table 2 we can observe the Cole-mak layout achieved the lowest cost in the 'Accessibility and Load' ergonomic criteria. Considering the frequency of letters in the English language, the most common characters include

| Keyboards | Cost for | | | | | |
|---|---|---|---|---|---|---|
| | Accessibility and load | Key Number | Hand Alteration | Consecutive finger use | Big steps | Hit direction |
| QWERTY | 1308 | 15571 | 13100 | 325 | 3826 | 3154 |
| AZERTY | 1555 | 15812 | 13398 | 346 | 3983 | 3272 |
| Colemak | 954* | 15571 | 12478 | 95* | 3780 | 3223 |
| Ant-keyboard | 971 | 15505* | 9915* | 390 | 1844 | 1689* |
| **LayoutRL** | 1553 | 15615 | 10722 | 703 | 1760* | 2722 |
| **MLRL** | 1060 | 15553 | 13241 | 668 | 2864 | 2842 |

Table 2: The individual cost comparison between the six ergonomic criteria for different keyboards and LayoutRL

E, A, R, and O. The placement of most of these characters in the Colemak layout approximates the hand's natural resting position on a keyboard. This leads to a lower cost for this ergonomic criteria. The cost for `Number of Keys` is more or less similar for all the layouts as it highly correlates to the number of alphabets that are to be typed. The slight change in values is caused by the use of 'shift' and 'alt' (in AZERTY) keys for typing capital alphabets and special characters. The `Hand Alteration` cost is lowest for Ant-keyboard, implying the possibility of using the same hand for consecutive characters while typing is minimal. Also, the low `Hit direction` cost indicates that while using the Ant-keyboard, the inner fingers are seldom used immediately after the use of an outer one. Colemak performs highly in the `Consecutive use of fingers` criteria, implying that the layout minimizes using the same finger of the same hand for back-to-back keystrokes. Though our generated keyboard may not perform exceptionally in specific ergonomic criteria except `Big step`, the cumulative cost of the layout outperforms all the traditional designs.

### 3.4 DISCUSSION

In this work, we demonstrate how Reinforcement learning can be used to optimize keyboard layout. Although various optimization algorithms have been used in this particular problem, we are the first to use the RL approach to solve for optimization. Our proposed method, 'LayoutRL' can optimize the keyboard layout and generate various optimized layouts over many standard keyboards such as 'QWERTY', 'Colemak', etc. Furthermore, the research on such optimization problems has indicated the popularity of 'QWERTY' and its familiarity with users. This led many researchers to focus on proposing layouts having close similarities to the 'QWERTY'. We adopted this constraint in our method also by dividing the state space into two sections for alphabets and non-alphabets. This 'QWERTY' like-similar properties addition to our algorithm enhances the usefulness of the best layouts from 'LayoutRL' because it enables these layouts to be deployed in real life to the users, not just from a theoretical perspective. Additionally, we address the ergonomic perspective considered in different keyboard optimization problems in this work. Six ergonomics criteria have been used in various keyboard layout optimization problems (Eggers et al., 2003; Deshwal & Deb, 2006). While LayoutRL optimizes the keyboard layouts utilizing these six criteria, we further explored these criteria by comparing their individual costs for different standard layouts. We believe this problem can be further optimized by incorporating aspects like unigrams, bigrams, sequence frequency of characters in the English language and ergonomic criteria along with their relative weights.

## 4 CONCLUSION

We introduced Reinforcement Learning (RL) for the first time to optimize layouts for English keyboards. Our approach integrates ergonomic criteria and practical design considerations to generate optimized layouts. The best layouts produced by our method demonstrate significant improvements over the traditional 'QWERTY' layout and other conventional designs. Our results indicate that layouts generated by 'LayoutRL' achieve approximately 12.4% lower costs than 'QWERTY'. Moreover, these layouts outperform conventional ones in terms of cost-efficiency when simulated with a standard English corpus. This RL-based optimization effectively generates highly efficient keyboard layouts and can be extended to various design contexts, including touchscreen devices, smartphones, virtual keyboards, and gesture typing. The broad applicability of our method highlights its robustness. In future work, we plan to explore the use of similar RL-based optimization techniques for keyboard layouts on a wide range of digital devices.

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
