# OpenReview forum: "LayoutRL: A Reinforcement Learning-Based Approach to Keyboard Layout Optimization"
_ICLR.cc/2025/Conference — ICLR 2025 Conference Withdrawn Submission_

### Official Review · Reviewer_MAAe · 2024-11-03

**Soundness:** 2
**Presentation:** 2
**Contribution:** 2
**Rating:** 3
**Confidence:** 3

**Summary:**

This paper formulates keyboard layout optimization as an RL problem, using heuristic objectives derived from prior work and several evaluation criteria and comparing with black box optimization-based baselines (e.g. Ant Colony optimization) along with standard layouts like QWERTY and Colemak.

**Strengths:**

I think formulating the problem this way requires some creativity (e.g. in the design of the state and action spaces), which I acknowledge and appreciate. I also acknowledge that the method appears to work reasonably well, as seen in the evaluation, and has been compared with a reasonable range of baselines. More broadly, I think the application of RL to problems in design and ergonomics is a very promising area and should be explored more, since RL has the potential to better capture behavioral factors of those who ergonomic systems are designed for.

**Weaknesses:**

My biggest concern with this paper is that the motivation for formulating keyboard layout optimization as an RL problem lacks sufficient motivation. Though I acknowledge RL is a powerful framework, keyboard layout optimization can be much more easily approached through other well-established methods such as those reviewed in this work (e.g. various black-box optimization algorithms).

The paper should clearly establish why RL provides unique advantages over traditional approaches for this particular problem. E.g. this could be justified if one or more of the following conditions were claimed and convincingly demonstrated:
- The goal is to learn user-adaptive strategies for keyboard layout design, where a generalizable policy could automatically customize layouts based on individual typing patterns, language preferences, usage contexts, etc.
- The policy can dynamically modify keyboard layouts over time in response to changes in user behavior or proficiency, shifts in their language usage patterns, different typing contexts (e.g., coding vs. stories), device form factors, etc. It would need to be established that there is a need for this, and users are capable of rapidly learning these layouts.
- The learned policy could transfer insights across languages and scripts, various input modalities (e.g. physical keyboards, touch screens, gesture input like swiping), new character sets or symbols, etc.

Without establishing these (or similar compelling motivations), RL seems to be a more complex solution to a problem that might be better served by simpler, more interpretable approaches.

In addition to this, I find it challenging to interpret the results. For example, the argument made (following from Table 1) is that LayoutRL gets the best results when considering keyboards that use the traditional layout structure. How can this be quantified, in terms of adherence to traditional layouts and the potential impact on downstream users? Without this, it’s difficult to understand what trade-offs are made when using Ant-keyboard, given its better overall results.

Another question is how difficult it would be to retrofit Ant-keyboard to adhere to the traditional layout structure, and what impact this would have on the evaluation. The search/optimization framework still seems like a more natural choice for this task, so it’s conceivable that if this could be done with a minor modification without substantially weakening the results, that might be preferable. A priori, it is not clear to me why the mixture of alphabet and non-alphabet keys compromises usability, or has other issues that arise from it. Additionally, plenty of optimization algorithms could likely admit such a straightforward constraint.

Finally, the adaptation from optimization to RL seems fairly direct: the objective terms described in 2.1 all come from Eggers et al. 2003, and the applicability of these various terms, etc. to RL contexts isn’t considered. As such, the knowledge contribution is limited; it would be interesting to know which of these are useful to an RL based approach, etc.

Minor notes and typos:
- Line 019: “an 12.4%” -> “a 12.4%”
- Line 025: “keyboard” -> “The keyboard”?
- Line 083: “combinational” -> “combinatorial”?
- The figure and table captions are too brief and uninformative. E.g. Figure 3 caption should clearly state what the two sides are, and provide a brief takeaway. Similarly, Table 2 caption should discuss the significance of the * notation used.

**Questions:**

What are the specific benefits obtained by modeling this problem using RL, over and above optimization based solutions? E.g. what scenarios does this solution apply in that optimization would not?

What are the (ideally, quantified or quantifiable) negative effects of non-adherence to the traditional layout structure’s separation of alphabet and non-alphabet keys?

How difficult would it be to modify Ant-keyboard or a similar approach to introduce the traditional layout constraint?

---

> ### Author Response · Authors · 2024-11-30
>
> Thank you for your review and queries regarding our work! We highly appreciate your thoughtful questions. We shall try to answer your inquiries to the best of our ability:
>
> 1) “The goal is to learn user-adaptive strategies for keyboard layout design, where a generalizable policy could automatically customize layouts based on individual typing patterns, language preferences, usage contexts, etc.
> The learned policy could transfer insights across languages and scripts, various input modalities (e.g. physical keyboards, touch screens, gesture input like swiping), new character sets or symbols, etc.
>
>     Without establishing these (or similar compelling motivations), RL seems to be a more complex solution to a problem that might be better served by simpler, more interpretable approaches.”
> - Thank you for your insightful comment!
> Yes, we agree that the points you raised in this comment need to be implemented for more enhanced usability.
> In our work, we focused on the general typing patterns and their common ergonomic criteria as the objective function. Furthermore, the state space in our RL algorithm can be exploited to fit the customized setup such as for touch screens, new character sets, or symbols. This can be performed by modifying our proposed architecture, however, we focused on the typical keyboard layouts in this work only. Furthermore, the language preferences can also be accommodated which we kept as a future work.
>
> 2) "I find it challenging to interpret the results. For example, the argument made (following from Table 1) is that LayoutRL gets the best results when considering keyboards that use the traditional layout structure. How can this be quantified, in terms of adherence to traditional layouts and the potential impact on downstream users? Without this, it’s difficult to understand what trade-offs are made when using Ant-keyboard, given its better overall results."
> - If you kindly look at Figure 4 (a,b) showing two samples of the best layout generated from the ‘LayoutRL’, we can observe that our generated layout has numbers and special characters together in the same way traditional layout has. However, the Ant-keyboard did not follow such restraints.
>
>   We agree that the quantifiable comparison between the traditional layout and our keyboard would reflect the ability of our method. For this, it is possible to propose a parameter that can be used to indicate how much any keyboard layout varies from the traditional layout structure.
>
> 3) " Another question is how difficult it would be to retrofit Ant-keyboard to adhere to the traditional layout structure, and what impact this would have on the evaluation. The search/optimization framework still seems like a more natural choice for this task, so it’s conceivable that if this could be done with a minor modification without substantially weakening the results, that might be preferable. A priori, it is not clear to me why the mixture of alphabet and non-alphabet keys compromises usability, or has other issues that arise from it. Additionally, plenty of optimization algorithms could likely admit such a straightforward constraint."
> - Yes, we agree that many algorithms can admit such constraints, however, these algorithms need to consider such constraints and train on the corpus repeatedly without learning any policy which may increase the complexity.

---

> > ### Comment · Reviewer_MAAe · 2024-11-30
> >
> > Thank you for your response. Unfortunately, I still don't think this constitutes a strong motivation for pursuing an RL approach to this problem. More specifically:
> > 1. The level of adaptation you describe could also be implemented using non-RL approaches, conceivably (such as the prior referenced work). This alone does not seem sufficient to motivate using RL.
> > 2. I understand this is a qualitative result, but since it is the key argument made in the paper, I think more convincing evidence to this effect is needed. Some way of quantifying this (e.g. a metric), along with a clear rationale (or ideally, some empirical evidence) for this metric correlating to downstream use, would be helpful.
> > 3. Again, it's not clear what the benefit of the policy is here since it is simply used once. Having strong evidence that this policy is useful for adaptation would be helpful, but this is not provided.
> >
> > I encourage you to consider these points when revising your work. While I think that, in principle, RL could be an interesting way to tackle this problem, its added complexity needs to be supported with a clear rationale and strong empirical evidence of its proposed benefits.

---

### Official Review · Reviewer_mgqk · 2024-11-04

**Soundness:** 2
**Presentation:** 2
**Contribution:** 1
**Rating:** 3
**Confidence:** 4

**Summary:**

The paper introduces LayoutRL, a reinforcement learning-based method for generating keyboard layouts that are optimized with regard to six ergonomic criteria taken from previous work. The authors motivate this by claiming that previous work has so far not evaluated ergonomic criteria sufficiently. In their results, they show that their method can design a keyboard layout that has a lower cost of corpus completion compared to some classical keyboard layouts and that scores better on the ergonomic criteria. In their comparison, they also compare against the optimized ant-keyboard and while their results show that their best layout scores worst on these metrics the authors reason that this is because they have restricted themselves to a QWERTY-like keyboard layout.

**Strengths:**

The method the authors introduce is novel in that no previous work has used reinforcement learning to optimize keyboard layouts. Optimizing keyboard layouts has the potential to improve human-computer interaction as it could improve the speed of typing, reduce the number of errors and improve the ergonomic features of a keyboard. In the past keyboards that improve upon QWERTY in this regard (i.e. most notably DVORAK) have seen some level of adoption. Their results show that their method can design keyboard layouts that show some improvement over some of the more classical layouts.

**Weaknesses:**

I have several concerns that must be addressed before this paper can be accepted:

- Limited novelty: Their novelty mostly stems from using an already known objective function [7] and optimizing it with a simple RL-based method. There is no distinct method contribution that would strongly suggest to include this submission in ICLR (that has a strong focus on algorithmic/method contributions).

- The writing should be improved (see the end) to make the paper easier to follow.

- The explanation of the method is not clear and should be rewritten or improved. Specifically:
  - Section 2 L106: What does it mean for an “alphabet” to be “mapped to each key”? Should that read as letters being assigned to keys?
  - Section 2.2.: It is unclear to me how these design choices were made. Previous work [2] used a dissimilarity metric to measure the distance of designed layouts from QWERTY instead of relying on hand-crafted design choices. More clearly stating why these design choices were made and how they compare to previous work would help to improve this section.
  - Section 2.2 L209: The authors mention the state space here which has not been explained or introduced yet.
  - Section 2.3: This section I find to be most confusing. The authors should provide an example to improve the section and reference Algorithm 1 earlier to guide the reader through their method.
  - Section 2.3: The authors should state the dimensions of the state/action value matrix. This would possibly help with some of the confusion. I assume it is of size 95 x 96?
  - Section 2.3: Why is the state transition probability non-deterministic? If the agent assigns a letter to a key it always results in the letter being assigned to this key, correct? This makes the state transitions deterministic.
  - Section 2.3.: Figure 2 shows B34 being mapped to G. What is B34 in this context? From Figure 1 it seems that it is not a coordinate in the keyboard layout.
  - Section 2.3.: Figure 2 (esp. the font size) needs to be larger. The text next to the matrix on the top left for instance is not readable at all on paper.
  - Section 2.4: The authors state that they run random episodes to randomize the state-action matrix for effective convergence but to me it reads more like an off-policy optimisation step depending on the inner workings of the adjust_corresponding_values function.
  - Section 2.4: In general the section could be improved by explaining why this algorithm was picked over more classical approaches. The authors could for instance explain why they picked this approach over a more simple epsilon-greedy learning approach for training their policy.

- Few comparisons: The authors should compare their best layout to other layouts in the literature, possibly the ones from [1,2,3,6]. While it seems to beat classical layouts - which were not optimized by any algorithm - their results indicate that other optimisation techniques might work better. I would personally also at the very least consider BZQ [6], GA-optimized keyboard [2], Carpalx Sim. Ann. (see [2]) and Dvorak. More would be better though.

- Evaluation of the designed layouts can be improved. First, previous work [7] has used a global score (a linear combination of the scores mentioned in the paper) that makes it easier to compare layouts by a single number. Second, other previous work [4] has calculated the Carpalx effort value [5] to compare keyboards. I suggest the authors include these values or other ways of quantifying their designed keyboard.

- Poor discussion and comparison to other techniques that optimize keyboards: The introduction motivates the paper by claiming that (in contrast to previous work) ergonomic criteria are not always modeled or validated and that keyboard optimisation methods should incorporate them. This seems to be correct but authors could have evaluated other optimized keyboard layouts later to show that there indeed is a gap in the literature. For instance by showing that while related work manages to produce optimized keyboard layouts that they score badly on their ergonomic criteria. This would provide a stronger motivation for both their method and the paper. Such a result could then also be reused in the introduction.

- Poor results: From Table 2 in the paper it seems that the final keyboard designed by LayoutRL is only the best in 1 out of 6 metrics. The ant-keyboard overall seems to be a better keyboard. This makes it unclear why LayoutRL should be used when ACO is available. The authors say that their keyboard is more similar to QWERTY and thus should be preferred but do not validate this claim with experiments. A user study might help to show that their keyboard is preferred by users over other optimised keyboards. In general though [2] has also accounted for similarity to QWERTY and the resulting keyboards thus need to be compared in such a study.

- In Table 2 it is unclear what MLRL is and where it comes from. It certainly is not mentioned in Table 1 and I cannot find another mention in the text. Also, why is it bold?

Other weaknesses:
- I found several spelling and grammar issues. For example L25 (The sentence start is missing), L226, L307 etc. This makes the paper hard to follow in places.
- I believe that throughout the work authors are misusing the word alphabet.

[1] Dunlop, Mark, and John Levine. "Multidimensional pareto optimization of touchscreen keyboards for speed, familiarity and improved spell checking." Proceedings of the SIGCHI Conference on Human Factors in Computing Systems. 2012.

[2] Khan, Ahmer, and Kalyanmoy Deb. "Optimizing keyboard configuration using single and multi-objective evolutionary algorithms." Proceedings of the Companion Conference on Genetic and Evolutionary Computation. 2023.

[3] Pradeepmon, T. G., Vinay V. Panicker, and R. Sridharan. "Hybrid estimation of distribution algorithms for solving a keyboard layout problem." Journal of Industrial and Production Engineering 35.6 (2018): 352-367.

[4] Nivasch, Keren, and Amos Azaria. "A deep genetic method for keyboard layout optimization." 2021 IEEE 33rd International Conference on Tools with Artificial Intelligence (ICTAI). IEEE, 2021.

[5] Krzywinski, M. "Carpalx keyboard layout optimizer." (2005).

[6] Fadel, Ali, et al. "QWERTY keyboard?}.? BZQ is better!." 2020 International Conference on Intelligent Data Science Technologies and Applications (IDSTA). IEEE, 2020.

[7] Eggers, Jan, et al. "Optimization of the keyboard arrangement problem using an ant colony algorithm." European Journal of Operational Research 148.3 (2003): 672-686.

**Questions:**

The motivation for using RL for optimizing keyboard layouts is not quite clear to me. Can the author elaborate on the motivation?

Will the authors provide code and trained model files to aid reproducibility?

---

> ### Author Response · Authors · 2024-11-29
>
> Thank you for your review and queries regarding our work! We highly appreciate your thoughtful questions. We shall try to answer your inquiries to the best of our ability:
> 1) "Q1. Section 2 L106: What does it mean for an “alphabet” to be “mapped to each key”? Should that read as letters being assigned to keys?"
> - Yes, It means the assignment of letters to keys. We understand it can be confusing as we used letters, symbols and alphabets interchangeably throughout the article.
>
> 2) "Section 2.3: The authors should state the dimensions of the state/action value matrix. This would possibly help with some of the confusion. I assume it is of size 95 x 96?"
> - The size of the matrix is 95 X 102. The number of keys is more than the number of letters because we consider ‘Shift + key’ as separate keys on the keyboard. The buttons like ‘enter’ and ‘backspace’ can have shift versions that are not being used in the QWERTY keyboard.
>
> 3) "Section 2.3: Why is the state transition probability non-deterministic? If the agent assigns a letter to a key it always results in the letter being assigned to this key, correct? This makes the state transitions deterministic."
> - In our version, the assignment of letters to a key is probabilistic, as the model has the option to assign a specific letter to any available keys on the keyboard. The probability of assigning certain key-letter pairs is updated after each episode after determining the cost of a layout.
>
> 4) "Section 2.3.: Figure 2 shows B34 being mapped to G. What is B34 in this context? From Figure 1 it seems that it is not a coordinate in the keyboard layout."
> - It means the letter ‘G’ is assigned to the 34th number Button in a particular epoch due to high probability.
>
> 5) "Section 2.4: The authors state that they run random episodes to randomize the state-action matrix for effective convergence but to me it reads more like an off-policy optimisation step depending on the inner workings of the adjust_corresponding_values function."
> - Yes, initially, we took an off-policy optimization step.
>
> 6) "In Table 2 it is unclear what MLRL is and where it comes from. It certainly is not mentioned in Table 1 and I cannot find another mention in the text. Also, why is it bold?"
> - Sorry, it was for another layout we generated and mistakenly added to the final submission. We are extremely sorry for this inclusion which we realised later just after the submission.
>
> 7) "The motivation for using RL for optimizing keyboard layouts is not quite clear to me. Can the author elaborate on the motivation?"
> - Optimization of keyboard layouts depends on the evaluation of the objective function which is dependent on a large corpus. The evaluation becomes much less computationally expensive compared to other optimization methods. In addition, the learned policy helps, often some sub-optimal sub-solutions may be used as an initial configuration, from which, the learned policies can lead to the local-optimal positions. This was the reason behind our motivation to use RL for optimizing keyboard layouts.
>
> 8) "Will the authors provide code and trained model files to aid reproducibility?"
> - Of course, if our work is accepted, we would gladly share our code for reproducibility.

---

> > ### Comment · Reviewer_mgqk · 2024-12-02
> >
> > Thanks for the clarifications. While some of the responses have been helpful,
> > I think most of the clarifications suggest that the paper needs more work to be
> > in a state in which it can be accepted at a conference such as ICLR.
> >
> > 1. I recommend to stick to one term.
> >
> > 2. Please make this more explicit in the paper.
> >
> > 3. The authors seem to misunderstand what the transition function refers to and
> > are confusing some of the fundamental aspects of RL. If assigning letter ‚B‘ to
> > the position B34 results in the letter ‚B‘ always being placed at position B34,
> > then the transition function of their environment is in fact deterministic.
> > This is independent of any considerations about how the model arrives at this
> > action. Compare this to an Atari agent that samples the action ‚right‘ in some
> > deterministic Atari game. If picking ‚right‘ always results in the agent going
> > ‚right‘ then this transition is deterministic even if the agent might sample
> > their actions at random.
> >
> > 4. Please make this more clear in your writing.
> >
> > Other major points I made regarding novelty, the evaluations, the poor
> > performance have sadly not been addressed.
> > Since the authors were unable to address my concerns, I prefer to keep my
> > score.

---

### Official Review · Reviewer_Coi6 · 2024-11-04

**Soundness:** 2
**Presentation:** 2
**Contribution:** 1
**Rating:** 3
**Confidence:** 3

**Summary:**

This paper presents a RL-based method to optimize keyboard layouts. Using a handcrafted reward function based on ergonomic principles and by training on large text corpuses, the proposed method is able to generate keyboard layouts with improvements to commonly used layouts (such as "QWERTY" keyboards).

**Strengths:**

There are a few strengths of this paper:

- Presents a reward function that takes into account important ergonomic principles and transforms them into actionable objective functions
- Generated Reasonable Optimized Keyboard Layout that follows traditional keyboard structure but with lower cost on large text corpuses
- Provided guidelines for restrictions on Keyboard Layout optimization problems that can be useful for further optimization

**Weaknesses:**

There are a number of significant weaknesses to this paper:

- **Unclear why RL-based approach is the best solution to this problem, or why this problem warrants an RL-based approach**: For a keyboard layout optimization problem with the author(s)' proposed formulation, the order of actions should not matter (given the return is evaluated at the end of the episode). It is thus unclear why a full policy should be learned to adapt to all state/action pairs, and why other optimization methods (such as simulated annealing) are insufficient for such a problem [1]. At the very least, the author(s) could include an ablation with these optimization methods that do not consider states.

- **Lack of use of recent innovation in RL**: Recent advancements in reinforcement learning have introduced more efficient learning algorithms (e.g., PPO), and it is unclear if the paper has cited or used them.

- **Missing Important Technical Details**: Some important technical details on the methods are missing from the paper. For instance, while the author(s) have presented elements of ergonomic rules that contribute to the 'objective function', it is unclear how exactly the final return function was formulated. Moreover, there are no details about the text corpus used to train the model, and I'd be curious to see whether the optimized keyboard layout can generalize to other text corpus in a different domain (e.g., trained on general domain corpus, evaluated on medical corpus).


References:

[1] N. Yang and A. D. Mali, "Modifying Keyboard Layout to Reduce Finger-Travel Distance," 2016 IEEE 28th International Conference on Tools with Artificial Intelligence (ICTAI), San Jose, CA, USA, 2016, pp. 165-168, doi: 10.1109/ICTAI.2016.0034.

**Questions:**

Directly corresponding to the weaknesses above:

- Can the author(s) clarify the rationale of using RL for the problem?
- Can the author(s) provide ablations against other optimization-based approaches?
- Did the author(s) use more advanced, recently introduced RL learning algorithms?
- Can the author(s) provide a formula for the final return function used?
- Can the author(s) provide details on the training text corpus?
- Was the evaluation done on a held-out text corpus? If not, would it be possible to provide experiment results on a held-out text corpus, preferrable in a slightly different domain than the training text corpus?

---

> ### Author Response · Authors · 2024-11-29
>
> Thank you for your review and queries regarding our work! We highly appreciate your thoughtful questions. We shall try to answer your inquiries to the best of our ability:
>
> 1) "Did the author(s) use more advanced, recently introduced RL learning algorithms?"
> - We used a very simplified version of the RL in our work. The motivation was to generate keyboard layouts with lower costs while using RL algorithm. We approached this problem as a simple RL (MDP) problem which gave promising results compared to other existing optimization methods, that’s why we did not explore more advanced RL methods.
> However, we do agree with the reviewer that a more advanced RL algorithm should be explored for such optimization which we intend to explore in our future works.
>
> 2) "Can the author(s) provide a formula for the final return function used?"
> - Yes, in section  2.1 (Keyboard Structure and Ergonomics Criteria Considered), we described six cost functions based on six ergonomic criteria: C1, C2, C3, C4, C5, C6 [equation 1-6]. The final return function i.e. cost for a particular layout is the weighted summation of these six costs which can be written as follows:
> the final cost function, f = 0.45C1+0.5 C2+
>  C3+0.8 C4 + 0.7C5 +0.6 C6
> The coefficients for each cost were taken from Eggers et al. (2003).
>
> 3) "Can the author(s) provide details on the training text corpus?"
> - Yes, the corpus was taken from this reference: https://www.corpusdata.org/formats.asp

---

> ### Author Response · Authors · 2024-11-29
>
> “Unclear why RL-based approach is the best solution to this problem, or why this problem warrants an RL-based approach: For a keyboard layout optimization problem with the author(s)' proposed formulation, the order of actions should not matter (given the return is evaluated at the end of the episode). It is thus unclear why a full policy should be learned to adapt to all state/action pairs, and why other optimization methods (such as simulated annealing) are insufficient for such a problem [1]. At the very least, the author(s) could include an ablation with these optimization methods that do not consider states.
> Can the author(s) clarify the rationale of using RL for the problem?"
>
> - Devising a proper cooling schedule for Simulated annealing to ensure optimized policy is a challenging task. Moreover, they are dependent on the evaluation of the objective function and as our objective function is dependent on a large corpus, the evaluation becomes much less computationally expensive compared to the simulated annealing or other metaheuristic formulation. In addition, the learned policy helps, often some sub-optimal sub-solutions may be used as an initial configuration, from which, the learned policies can lead to the local-optimal positions more efficiently.

---

### Official Review · Reviewer_Tr4n · 2024-11-09

**Soundness:** 2
**Presentation:** 1
**Contribution:** 2
**Rating:** 3
**Confidence:** 4

**Summary:**

This work proposed a RL base method to find the best keyboard layout specifically designed for English. The proposed framework aims to optimize the ergonomic parameters/variables, and the authors have shown that the layout identified by their RL algorithm achieves significant improvement over traditional keyboard layout.

**Strengths:**

1. RL algorithm for finding the optimal keyboard layout is a new angle to approach the problem.

2. In the fields of Human Computer Interaction, the paper holds great potential where a lower cost layout will not only improve user experience, but also improve productivity. From that perspective, the high level direction/scope of finding a more optimized/improved keyboard layout holds significant importance. However, I have questions about the significance of this particular work in the absence of rigorous validation or user study.

3. The reduction in cost of the layout detected by the RL algorithm is statistically significant compared to the cost of traditional layout, which proves that the RL framework is generally capable to identifying the best layout from a massive corpus, hence eliminating human effort to manually think of and design a better layout.

**Weaknesses:**

1. While I am excited about the high level direction (optimizing the keyboard layout with ergonomics and productivity considerations), the paper does not explain the implications and significance of the results. For example, how much performance and productivity does the new layout provides (e.g., speed of typing) for actual human participants? If a participant is already trained with a QWERTY layout, how much performance benefit can we expect to see? How long does it take for a person to get used to the new layout (assuming that the person is already trained with QWERTY)? What are the longer term kinematics benefits? In order to actually evaluate success of the project, one needs to be able to answer/discuss some of these points. Otherwise, it's very difficult to understand the significance of the work.

2.  Why the RL approach is better than a standard optimization based approach? For example, why not genetic search with the same set of constraints? The paper does not provide detailed comparison with the proposed approach (that convert the optimization problem to a sequential decision making RL problem) to a large body of standard optimization based approaches?

3. The performance in the table isn’t highlighted, for which is hard for readers to realize the significant improvement at the 1st time. Although the cost of layout identified by RL is not the best, a gradient table cell color identifying 1st, 2nd, and 3rd best performance will improve the table presentation.

4. The quality of figures and tables can be improved. Some of the text on the figures can not be easily read.

**Questions:**

how much performance and productivity does the new layout provides (e.g., speed of typing) for actual human participants? If a participant is already trained with a QWERTY layout, how much performance benefit can we expect to see? How long does it take for a person to get used to the new layout (assuming that the person is already trained with QWERTY)? What are the longer term kinematics benefits? Why the RL approach is better than a standard optimization based approach? For example, why not genetic search with the same set of constraints?

---

> ### Author Response · Authors · 2024-11-29
>
> Thank you for your review and queries regarding our work! We highly appreciate your thoughtful questions. We shall try to answer your inquiries to the best of our ability:
> 1) " how much performance and productivity does the new layout provides (e.g., speed of typing) for actual human participants?"
> -
> Sorry, but we did not include any human participants in this work. Rather, we considered the RL agent to be mimicking the actions while considering certain ergonomic criteria. While we agree that human participants would definitely enhance the evaluation of this work as well as acceptability, integrating more ergonomic criteria into the system can become an alternative for exploring such evaluations.
>
> 2)  Why the RL approach is better than a standard optimization based approach? For example, why not genetic search with the same set of constraints?
> - RL Approach has the ability to maximize the objective function considering a long-term reward system whereas other approaches do not consider such a long-term scenario. Additionally, RL agents can learn about the environment through iterative training. For example, our approach assigns the alphabet to the respective key sequentially, and then once all the alphabets are assigned, the cost is calculated which is considered to be a penalty system, with higher cost indicating some state/actions are going to be penalized. Such long-term consideration of the optimization technique is often missing in other approaches, where other usually considers random mutations. As you already mentioned genetic search that follows a random approach rather than considering the effect of the sequential actions in each scenario.
>    Furthermore, RL based approach can adapt to changes in the environment over time which gives such an approach to be used in optimizing other language keyboards.
>
>
>    Eggers et al. (2003) used Ant Colony Optimization which also lacks the inherent learning of the keyboard state space and can struggle with complicated language keyboards such as Bengali, Hindi, or Arabic. However, our approach has the ability for such adaptations which we intend to show in our future works. Furthermore, we demonstrated that even though Ant Colony Optimization should achieve global optimization, ours performed better in achieving the optimal value.

---

> ### Author Response · Authors · 2024-11-29
>
> 1) "The performance in the table isn’t highlighted, for which is hard for readers to realize the significant improvement at the 1st time. Although the cost of layout identified by RL is not the best, a gradient table cell color identifying 1st, 2nd, and 3rd best performance will improve the table presentation."
> - You are absolutely right in identifying this visualization problem in this table!

---

### Note · Authors · 2025-01-23

I have read and agree with the venue's withdrawal policy on behalf of myself and my co-authors.